# Collective Immunity to the Measles, Mumps, and Rubella Viruses in the Kyrgyz Population

**DOI:** 10.3390/vaccines13030249

**Published:** 2025-02-27

**Authors:** Anna Yurievna Popova, Viacheslav Sergeevich Smirnov, Svetlana Alexandrovna Egorova, Zuridin Sharipovich Nurmatov, Angelika Marsovna Milichkina, Irina Victorovna Drozd, Gulzada Saparbekovna Dadanova, Gulnara Dzhumadylovna Zhumagulova, Ekaterina Mikhailovna Danilova, Zharkynbek Orozbekovich Kasymbekov, Victoria Georgievna Drobyshevskaya, Gulsunay Zhumabaevna Sattarova, Oyuna Bayarovna Zhimbaeva, Edward Smith Ramsay, Zhanylay Nuridinovna Nuridinova, Valery Andreevich Ivanov, Altynai Keneshpekovna Urmanbetova, Areg Artemovich Totolian

**Affiliations:** 1Federal Service for the Oversight of Consumer Protection and Welfare, 127994 Moscow, Russia; 2Saint Petersburg Pasteur Institute, 197101 St. Petersburg, Russiatotolian@spbraaci.ru (A.A.T.); 3National Scientific and Practical Center for Control of Viral Infections, National Institute of Public Health, Bishkek 720005, Kyrgyzstan; 4National Center for Immunoprophylaxis, Bishkek 720005, Kyrgyzstangulnara.zhumagulova@privivka.kg (G.D.Z.);; 5National Institute of Public Health Ministry of Health of the Kyrgyz Republic, Bishkek 720005, Kyrgyzstan

**Keywords:** collective immunity, measles, mumps, rubella, seroprevalence, antibodies, Kyrgyz Republic, population, cohort study

## Abstract

Specific prevention of measles, mumps, and rubella (MMR) is the main prerequisite for a radical reduction in the incidence of these infections in the Kyrgyz Republic (KR). An increase in the number of seronegative individuals observed in recent years has led to an increase in measles incidence. This is directly related to a decrease in collective immunity, which provides protection for the population only in conditions of a high density of immunized individuals and their uniform distribution in the population. The only way to estimate the number of immunized individuals is by conducting serological surveys of collective immunity. Aim of the study: The aim was to study the level of collective immunity to the MMR viruses in the KR. Materials and methods: This study involved a cohort of 6617 residents (volunteers) aged 1 to 70^+^ years, formed in accordance with the Rospotrebnadzor program “Assessment of collective immunity to vaccine-preventable and other relevant infections” and stratified by age and region of residence. During the study, participants filled out a questionnaire and gave venous blood samples to determine IgG antibodies to MMR viruses (ELISA using certified Russian test systems). Results: In December 2023, collective immunity ensured epidemiological well-being only with respect to rubella. The volunteer seropositivity was 94.2% (95% CI: 93.7–94.8). The average measles seropositivity was 78.9% (95% CI: 77.9–79.9). It was significantly lower in children aged 1–17 years and significantly higher than the cohort average in individuals aged ≥18 years. The average mumps seropositivity was 76.4% (95% CI: 75.3–77.4), with the minimum level noted among individuals aged 12–29 years (63.8%; 95% CI: 61.4–66.2). The maximum levels of mumps seropositivity were noted among children aged 6–11 years and older adults who had likely experienced mumps (50–70^+^ years). Seroprevalence distributions by activity correlated with age distributions (all infections). Conclusions: The system of targeted prophylaxis of vaccine-preventable infections adopted in the KR has contributed to the formation of a high level of rubella collective immunity and, to a lesser extent, measles and mumps immunity. The recent trend towards increases in measles and mumps incidence in the KR requires additional efforts to increase collective immunity to these infections.

## 1. Introduction

During the novel coronavirus pandemic, the state of collective immunity to COVID-19 has been of particular importance [1]. In addition, immune control over so-called “vaccine-preventable infections”, such as measles, mumps, and rubella, continues to be relevant [2]. To solve these urgent problems, specialists from the St. Petersburg Pasteur Institute developed a system for monitoring collective immunity formed in response to current viral infections [2,3,4].

It is well known that measles, mumps, and rubella (MMR) are highly contagious viral illnesses that continue to pose serious public health problems. In addition to the obvious demographic aspects, vulnerability to measles and mumps is sometimes elevated for other reasons. These can include transient immunosuppression resulting from other viral infections, or any number of physiological imbalances that might exist in various organs and systems when serious medical problems are present [5,6].

The key approach to preventing these infections has been the widespread introduction of mass immunization with vaccines, which has reduced the incidence of these illnesses to sporadic levels in many countries [7,8]. Unfortunately, despite significant efforts by the medical community, complete eradication of MMR on a global scale has not yet been achieved. These infections occur in almost all regions globally, and relative well-being is sometimes perturbed by increases in morbidity. According to “Our World in Data”, 205,101 measles cases were registered globally in 2022 (the most recent published data) [9]. Rubella cases were significantly lower (17,865 in 2022). It was not possible to find global data on mumps. However, according to the Annual Epidemiological Report (2022), 2593 mumps cases were registered in the European Union in 2022 (0.7 per 100K pop.) [10]. It should be noted that many countries in Africa and Southeast Asia do not keep track of rubella or mumps incidence, perhaps considering them mild illnesses that do not require any medical response.

In addition, several serious factors have an important influence on parents’ decision to vaccinate their children or not. These are as follows: (1) a negative attitude towards vaccination due to propaganda, pseudoscientific information, or previous experience of adverse events; (2) religious or cultural factors; (3) low socioeconomic status, income, or education; (4) reliance on alternative or non-traditional medicine; and/or (5) a large number of children in the family [11,12,13,14].

The first three factors are the most significant and difficult to correct. Often, the population has to be convinced of the efficacy and safety of vaccination, including against MMR. An important role in this process is played by the media and health workers. In countries with a Muslim majority, the corresponding positions of religious leaders are also influential. Unfortunately, even the resolution of these issues does not yet guarantee the complete elimination of MMR without meeting other requirements. These include the successful production and delivery of vaccines to all countries globally and ensuring sufficient coverage of the population (≥92–95%) with preventive MMR vaccination [11,15].

The most important component of all programs for the control and elimination of vaccine-preventable infections is the monitoring of collective immunity. It is well known that insufficient levels (assessed primarily as specific Ab levels) can lead to the accumulation of susceptible individuals and an increased likelihood of outbreaks [16]. Researchers from most countries globally continuously monitor MMR seroprevalence. For example, a 2014 study of the average share of seropositive individuals in PRC found 81.1% (95% CI: 78.0–83.9) for measles, 65.6% (95% CI: 62.2–69.4) for rubella, and 63.2% (95% CI: 59.4–66.8) for mumps. In that situation, insufficient collective immunity levels required additional vaccination [15].

In Germany, a similar study (7115 volunteers) indicated low MMR outbreak risk in 2021 due to high seropositivity: 89.9% for measles, 94.0% for rubella, and 84.2% for mumps [11]. An authoritative source of information, the Cochrane Review, describes the following findings regarding the effectiveness of vaccination. The measles vaccine reached 95% (survey of 12,000 children). The rubella vaccine reached 89%, increasing over time to 95% (survey of 2279 children). The effectiveness of mumps vaccination was 72% (survey of 9915 children) [17].

More than 21 years of experience using the trivalent M-M-R II vaccine for specific prophylaxis has shown that even a single dose forms adaptive immunity among recipients to measles in 87.4–100%, to rubella in 90.0–100%, and to mumps in 79.5–100% [18]. These data indicate the ability of MMR vaccines to form a long-term, stable immune response. Accordingly, the level of collective immunity formed can vary depending on a number of the aforementioned, mainly socioeconomic, factors. Taking this into account, the authors set the goal of assessing the state of collective immunity to MMR pathogens in the population of the Kyrgyz Republic (KR).

## 2. Materials and Methods

### 2.1. Characteristics of the Surveyed Volunteer Cohort

To conduct a cross-sectional, randomized study of collective immunity in the Kyrgyz population in November 2023, a representative cohort of volunteers was formed from residents across the Republic, randomized by age and regional characteristics. The study was conducted under the Rospotrebnadzor program “Assessment of collective immunity to vaccine-preventable and other relevant infections” as part of the implementation of the Decree of the Government of the Russian Federation (dated 18 April 2023 No. 972-p). The study plan was approved by the local ethics committees at the Saint Petersburg Pasteur Institute and the “Preventive Medicine” Scientific and Production Association, Kyrgyz Ministry of Health. Before initiation, all participants, or their legal representatives, were familiarized with the purpose and methodology of the study and signed informed consent.

Random selection of volunteers was carried out by online questionnaire with randomization by age and region using a web application. The exclusion criterion was presence of any active infectious illness (any etiology) [4,19]. The total size of a representative sample was calculated using an online calculator created on the basis of the local De Moivre–Laplace theorem [20]. The volunteer cohort consisted of 6617 people and included 9 age groups (in years): 1–5, 6–11, 12–17, 18–29, 30–39, 40–49, 50–59, 69–69, and 70^+^ (Table 1).

Age groups varied from 407 to 1025 individuals (Table 1). The smallest share of volunteers was in the group aged ≥70 years, hereafter shown as “70^+^”. The largest group was children aged 6–11 years. The distribution of the volunteer cohort by age was generally representative of the demographic structure of the Kyrgyz population. In the population, children under 7 years old make up 17%, and adolescents aged 7–17 years constitute 21%. The smallest share of the population belongs to the age group of 60–70^+^ years. Overall, the ratio of males to females was 28.8%:71.2% (1:2.5), with some variation between age groups. In children’s age groups, the percentages of males and females were comparable. In adult groups, women significantly exceeded men.

The surveyed cohort included volunteers from all seven Kyrgyz regions and two federal cities (Bishkek, Osh) (Table 2).

The largest number of volunteers lived in the Osh and Jalal-Abad regions (Table 2). As expected, residents of the capital, Bishkek, took an active part in the study. The share of city residents significantly exceeded the share of volunteers from the Chui region, where the capital is located. The smallest shares of volunteers were from the Osh and Talas regions. The gender structure generally corresponded to that presented in Table 1 without any notable features.

The distribution of volunteers by activity showed that the study generated interest among people of different professions, although to varying degrees. To increase the statistical significance of the results, small volunteer categories (artists, military personnel, tourism industry specialists) were combined with similar categories (scientists, civil servants, and “other”, respectively). In addition, workers in manufacturing and transportation were combined into one group. Schoolchildren and medical workers participated most actively (Table 3).

### 2.2. Research Methods

Volunteers filled out a questionnaire providing relevant personal information, including information on presence of chronic diseases, blood transfusions, surgeries, MMR history, and MMR immunization (vaccination/revaccination). Immune status was assessed by determination of specific antibodies (Ab) to MMR viruses. For this, blood samples were collected from all volunteers (peripheral veinous draw into EDTA vacuum tubes). After centrifugation, plasma was separated from cellular elements and stored until analysis (+4 °C). Commercial ELISA reagent kits (Vector-Best, Russia) were used to profile IgG antibodies against the corresponding viruses in volunteer blood plasma: “VectoMeasles-IgG” (quantitative, qualitative) for measles virus, “VectoRubella-IgG” (quantitative, qualitative) for rubella virus, and “VectoMumps-IgG” (qualitative) for mumps virus. In all cases, analysis was carried out according to manufacturer instructions.

When determining anti-measles IgG, classification was as follows: positive when ≥0.18 IU/mL, negative when <0.12 IU/mL, and “gray zone” when 0.12–0.17 IU/mL. When determining anti-rubella IgG, the result was considered positive when ≥10 IU/mL and negative when <10 IU/mL. When determining anti-mumps IgG, a cutoff (ODcrit. = ODaverage for K- + 0.3) and a positivity coefficient (PC = OD sample/ODcrit.) were calculated. The results were classified accordingly: positive when PC > 1, negative when PC < 0.8, and inconclusive when 0.8 ≤ PC ≤ 1.

Statistical processing was carried out using the methods of variation statistics in software (Excel 2011). Statistical analysis of shares was carried out according to the method of Wald and Wolfowitz [21], as modified by Agresti and Coull [22]. Calculation of the significance of differences in shares was carried out using the z-test. To assess the reliability of differences in the compared indicators, a probability level of *p* = 0.05 (or better) was used, unless otherwise stated.

## 3. Results

### 3.1. Measles

#### 3.1.1. Epidemiological Situation

The situation with measles in the Republic can be classified as difficult. According to official data of the Kyrgyz Ministry of Health, one case was registered in the group “children aged 1–14 years” in 2022. However, in 2023, there were already 7028 cases, including 4839 among children. The overall intensity in the Republic was 74.2 per 100 K population. Among children aged 1–14 years, it was 211 per 100 K. In the first 8 weeks of 2023, only isolated measles cases were reported. However, starting from the eighth week onwards, sporadic reports were replaced by a continuous increase in cases (Figure 1). Over 6 months in 2024, out of 12,184 individuals with measles, 9994 were hospitalized, and six cases were fatal.

Incidence was described by a complex fourth-degree polynomial in which two sections can be distinguished. The first section (flat) occurred between weeks 8 and 28. It was characterized by a slight increase in incidence, approximated by a linear regression of the following form: y = 5.39x + 6.60 (R^2^ = 0.62; *ρ* = 0.788; *p* = 0.001; slope (k) = 5.22). Starting from week 28, incidence increased noticeably, and the regression curve acquired the shape of a bell-shaped curve shifted to the right. This was described by a third-degree regression of the following form: y = 0.004x^3^ − 1.30x^2^ + 49.81x + 55.45 (R^2^ = 0.783; *ρ* = 0.524; *p* = 0.001). The ascending section characterizes a slow increase in incidence from week 29 (2023) to week 5 (2024) with an angular coefficient of k = 11.2. The trend then changes sign, and a rapid decrease in measles incidence begins, from a rate of k = 26.5 to a minimum at week 24 (2024).

Presumably, the increase in cases was accompanied by an increase in immune coverage, which led to a change in trend and an increase in convalescents. Obviously, the highly contagiousness nature of measles affected incidence dynamics in all Kyrgyz age groups and regions. Expectedly, the highest case numbers were identified in the most populated regions: Bishkek city (Chui region) and Chui region itself. The lowest numbers were in the predominantly mountainous regions of Talas (Chatkal range), Naryn (Tian Shan Mountain system), and Issyk-Kul (Kyrgyz Ala-Too) (Figure 2). The latter are regions featuring the smallest populations (Table 2).

In terms of age, the highest number of illnesses was identified among young children: 26.0% (95% CI: 25.0–27.0) in those <1 year old, 43.7% (95% CI: 42.5–44.9) in those aged 1–4 years, and 18.1% (95% CI: 17.2–19.0) in those aged 5–9 years. The total contribution of other age groups to prevalence was 12.3% (95% CI: 11.4–12.6). All differences were significant (*p* < 0.001). In light of such high incidence, it was important to understand the structure of patient vaccination. For this purpose, 12,133 individuals under 30 years of age who had experienced measles in 2024 were analyzed. The choice of this age range was due to the fact that measles most often occurs in children aged 1 to 3 years, progressively declining with advancing age (Figure 2). The maximum number of children who contracted measles under the age of 1 year was noted in three regions (Jalal-Abad, Osh, Issyk-Kul) and Bishkek city. In the remaining Kyrgyz regions, measles was more often registered in the age group “1–4 yrs” and significantly less in the group “5–9 yrs”.

To better understand such high incidence, the vaccination status of those infected was analyzed (Figure 3).

Among those who became ill, the share of individuals fully vaccinated against measles (2 doses) was 2.7% (95% CI: 2.4–3.0). The share of those vaccinated with the first dose was 5.1% (95% CI: 4.7–5.5), and the differences were significant (*p* < 0.05). The largest number of ill (71.2%) was identified among those unvaccinated either due to age or refusal to vaccinate. The share of those not vaccinated for other reasons was 21%. These data convincingly indicate that the main factor behind measles cases is the refusal to vaccinate.

Considering the fact that the majority of those infected were children, it was the pediatric population that comprised the majority of the groups presented in Figure 3. The “unvaccinated due to age” group was represented exclusively by children under 1 year of age. Among the other groups (vaccinated 1 or 2 doses, medical exemption, migrants, refusal of vaccination), representation was mainly children aged 1–14 years and adolescents aged 15–19 years, with adults constituting only isolated cases. Adults predominated in the “other reasons” group in which background factors could not be determined.

Thus, the presented data showed that the main prerequisites and sources of measles spread among the population remain individuals who have not received proper vaccination and associated low, or absent, adaptive immunity. In this regard, there was a need to assess immunity to the measles virus and the vaccination structure using a cohort of volunteers representative of the population as an example.

#### 3.1.2. Structure of Measles Collective Immunity

In light of the worsening epidemic situation with measles, the state of collective immunity in its various aspects is of particular importance. Foremost, specific immunity in different age groups is a key factor. The average measles seroprevalence in the Kyrgyz population was 78.9% (95% CI: 77.9–79.9). Seropositivity among children aged 1–17 was significantly lower than the cohort average at 60.7–66.0%. Seropositivity among adults (18–70^+^) was significantly higher than the cohort average at 85.2–95.0% (Figure 4).

Taking into account that the cohort was represented by volunteers from across the Republic, it seemed appropriate to also evaluate the distribution of seroprevalence by region (Appendix A).

In general, the distribution of seroprevalence by region was almost uniform. Significant differences were noted in four regions. In Osh city and the Chui region, values were significantly higher than the Republic overall (91.8%, 95% CI: 87.8–94.8 and 82.6%, 95% CI: 80.0–85.1). In the Batken and Jalal-Abad regions, they were significantly lower (69.1%, 95% CI: 65.1–72.9 and 74.9%, 95% CI: 72.3–77.3) (Appendix A).

An analysis of seroprevalence by activity type correlated with the previously noted relationship between seroprevalence level and age. Significantly lower seroprevalence was shown in young people (1–17 years), such as preschoolers (68.5%, 95% CI: 64.8–72.1) and schoolchildren (59.2%, 95% CI: 56.8–61.6). The highest seroprevalence > 87%, significantly different from the cohort average, was observed in adult volunteer categories: medical professionals, scientists and artists, civil servants, and pensioners (Appendix A).

In addition to qualitative assessment, quantitative anti-measles antibody levels were assessed. An analysis of the data presented in Appendix A showed that the smallest numbers of volunteers were in the groups with antibody levels of 0.51–1.0 IU/mL and >2.0 IU/mL. The largest numbers were in the groups with antibody levels of 0.18–0.5 IU/mL or seronegative (<0.18 IU/mL). Further analysis revealed a relationship between Ab levels and age (Figure 5).

Titer trends by age interval were complex and described by third-degree polynomials. It is noteworthy that the regression curves for seronegative volunteers (<0.18 IU/mL) and individuals with maximum Ab levels (>2.0 IU/mL) are opposites in form. The seronegative curve mainly moves downward across advancing age groups, with a slope of about −6.0 in the central region (the straight section between intervals “6–11 years” and “4–49 years”).

In contrast, the regression curve for maximum titers (>2.0 IU/mL) featured an upward trend (slope ~ 5.5). It is worth noting that the shape of the regression curves is not accidentally described by third-degree polynomials. Spearman’s correlation analysis showed satisfactory agreement between the theoretical curve and the actual distribution of experimental points. The *ρ* values varied from 0.90–0.91 (*p* < 0.001) for the two described trends. As for the remaining curves, the trend for the 0.18–0.5 IU/mL interval followed the seronegative trend, albeit with a smaller slope. The trend for the 1.01–2.0 IU/mL interval was almost parallel to that of the maximum Ab interval, as indicated by identical R^2^ indices and close values (*ρ*, *p*) (Figure 5).

Among children, young adults, and middle-aged adults (≤40 years), individuals with low Ab levels (0.18–0.5 IU/mL) predominated, probably acquired through vaccination. Among adults > 40 years, seropositive individuals predominantly had high (1.0–2.0 IU/mL) or very high (>2.0 IU/mL) Ab levels, perhaps due to childhood illness.

The surveyed cohort was heterogeneous in terms of history (infectious, vaccinal status). Information was available for 5530 volunteers. Depending on their history, the entire cohort was divided into four groups: group 1—“sick, never vaccinated” (SNV, n = 59), group 2—“sick, vaccinated” (SV, n = 48), group 3—“never sick, never vaccinated” (NSNV, n = 1811), and group 4—“never sick, vaccinated” (NSV, n = 3612).

As the data shows, the majority of volunteers (65.3%; 95% CI: 64.1–66.6) were “never sick, vaccinated” (NSV group) (Figure 6). Such individuals prevailed in almost all age groups, and that status was the largest among children (73.9–79.7%). In adults, high percentages of “never sick, never vaccinated” were noted: from 35.8% among younger individuals (aged 18–29) to 54.1% in the elderly (aged 70^+^). This may be associated with vaccination policies. Volunteers who had experienced overt measles and reported the illness in the questionnaire accounted for 2%. Of them, 1.1% (95% CI: 0.8–1.4) were not vaccinated, and 0.9% (95% CI: 0.7–1.1) were vaccinated.

An analysis of seroprevalence and IgG levels was conducted by volunteer history (Appendix A, Figure 7 and Figure 8). Due to the small number of volunteers with overt measles history (SNV, SV), it was not possible to assess the reliability of differences in these groups. However, seroprevalence was higher in these groups than among volunteers who denied experiencing illness. The values were as follows: 84.7% (95% CI: 73.5–91.8) for SNV and 93.8% (95% CI: 83.2–97.9) for SV. Relatively high Ab levels (≥1.0 IU/mL) were also a feature of the SNV and SV groups (Figure 7).

Among volunteers without a history of overt measles, seropositive individuals were as follows: 83.4% (95% CI: 81.6–85.0) in the NSNV group, and 75.4% (95% CI: 74.0–76.8) in the NSV group. Among such seropositive individuals, low and medium Ab levels (<1.0 IU/mL) predominated.

### 3.2. Rubella

#### 3.2.1. Epidemiological Situation

The epidemiological situation regarding rubella can be characterized by two main features. On the one hand, there was some increase in sporadic morbidity. On the other hand, overall rubella cases are not numerous enough to exacerbate conditions. According to official Kyrgyz Ministry of Health data, only one case was detected in the pediatric population in 2022. In 2023, 11 cases were already registered. The prevalence was 0.2 cases per 100 K population. In children aged 0–14 years (among whom all cases were registered), the prevalence reached 0.5 per 100 K population (Table 4). In the last ten years, no cases of congenital rubella syndrome have been registered in the Republic.

Of course, there is no reason to expect a mass, or even noticeable, outbreak at such levels. From this standpoint, the rubella situation can be considered quite favorable.

Mass vaccination is actively implemented in the KR, most often with a trivalent live MMR vaccine. Despite opposition from individual parents, overall vaccination coverage of the Kyrgyz population reached 60% in 2023. The vaccine is used for scheduled immunization of children aged 1 year, with revaccination at 6 years. In this context, there are sufficient grounds to believe that the active vaccination campaign has provided the basis for the formation of strong collective immunity to rubella among the population.

#### 3.2.2. Structure of Rubella Collective Immunity

The data obtained confirmed a high level of seroprevalence among volunteers of all age categories. The cohort average was 94.2% (95% CI: 93.7–94.8). The lowest level was noted in children aged 1–5 years. The highest was in the group 40–49 years. Graphical representation (seroprevalence by age) also confirmed the conclusion that high levels of anti-rubella humoral immunity are present (Figure 8).

Of course, with such seroprevalence, there is no reason to expect a significant aggravation of rubella epidemiology, insofar as seronegative individuals, even in the most “unfavorable” age groups, do not exceed 9%. In principle, the aforementioned analysis was conclusive, yet it seemed advisable to consider all aspects.

When comparing the values by region and field of activity, no significant differences were found (Appendix A).

Figure 9 presents the quantitative aspects of rubella seroprevalence in different age groups. The majority of seropositive volunteers, regardless of age, had anti-rubella Ab levels of 25–100 IU/mL (47.4%; 95% CI: 46.2–48.6). High Ab levels (>100 IU/mL) were typical mainly for individuals aged ≥30 years.

An analysis of titer trends by increasing age interval showed a number of features specific only to rubella. The groups < 10 IU/mL and 10.1–25 IU/mL were characterized by a weak, downward linear trend in the direction from 1–5 years to 70^+^. The trends were largely random, as evidenced by insignificant coefficient values (*ρ*) and low determination coefficients (R^2^).

The largest number of volunteers had Abs in the range of 25–100 IU/mL. The regression curve in this case was described by a third-degree polynomial with a statistically insignificant correlation coefficient. The two remaining levels (100.1–200 IU/mL, >200 IU/mL) were also described by third-degree polynomials with insignificant correlation coefficients.

As with measles collective immunity, the surveyed cohort was heterogeneous in terms of history (infectious and vaccinal status) (Figure 10). Information was provided by 5544 volunteers. Based on history, the entire cohort was divided into four groups: group 1 (SNV, n = 27); group 2 (SV, n = 21); group 3 (NSNV, 2316); and group 4 (NSV, n = 3180).

The structure of rubella history (infectious, vaccinal) was in many ways similar to that of measles (Figure 11). The number of volunteers who had experienced manifest illness (SNV, SV) was insignificant. All age groups were represented almost exclusively by two groups, NSNV and NSV, in different ratios depending on age. In children’s groups, NSV exceeded 65%. Among those aged 18 years and older, NSNV increased from 42.3 to 58.6%.

An analysis of seroprevalence in the groups did not reveal significant differences, especially considering the small number of volunteers who had experienced rubella (Appendix A). The specific group values were as follows: 88.9% (95% CI: 71.9–96.1) among SNV, 95.2% (95% CI: 77.3–99.2) among SV, 93.9% (95% CI: 92.9–94.8) among NSNV, and 95.0% (95% CI: 94.2–95.7) among NSV. An evaluation of anti-rubella IgG levels showed that, regardless of history, volunteers with medium (25.1–100 IU/mL) or high (>100 IU/mL) titers prevailed (Figure 11).

### 3.3. Mumps

#### 3.3.1. Epidemiological Situation

The epidemiological situation for mumps in the KR can be assessed as stable. In 2022, 167 cases were detected. In the first 11 months of 2023, 162 cases were registered (Table 5).

Sporadic cases of mumps were registered in all regions, and the largest numbers were in Bishkek and the Chui region. The Naryn region was the most favorable, wherein only one case was detected. The intensity for the Republic was 2.3 cases per 100 K population. Therefore, the incidence can be assessed as favorable. An additional positive fact, to a certain extent, that can be considered is the frequently self-limiting course of mumps. In about 20% of cases, it is not accompanied by noticeable clinical manifestations and remains unrecognized [23].

#### 3.3.2. Structure of Mumps Collective Immunity

The average mumps seroprevalence in the volunteer cohort was 76.4% (95% CI: 75.3–77.4). This is lower than the value for rubella but comparable to the value determined for measles. Seroprevalence in different age groups was characterized by noticeable heterogeneity. High seropositivity was noted in children aged 6–11 years (82.0%; 95% CI: 79.6–84.4) and adults aged 50–70^+^ years (>80%). Lower values were observed among individuals aged 12–29 years, not even reaching 70% (Figure 12).

It should be noted that seroprevalence was almost uniform across all regions without any significant differences, with one exception. In the Issyk-Kul region, the value (67.3%; 95% CI: 63.1–71.2) was below the national average (Appendix A).

Different results were obtained when assessing seroprevalence by activity. It was heterogeneous, yet showed similarities to age-associated seroprevalence patterns (Appendix A). The lowest seropositivity was noted in the student group, where it was significantly lower than the cohort average (50.6%; 95% CI: 42.7–58.5; *p* < 0.0001). This group mainly included individuals aged 16–29. A significantly higher value was observed in pensioners (85.6%; 95% CI: 82.6–88.3; *p* < 0.0001), and this is also explained by the age composition of the group (those >60 years).

In conclusion of the section, we consider seroprevalence levels and age distribution depending on history. Information on infectious and vaccinal status was provided by 5517 volunteers (Figure 13). As the data show (Appendix A), individuals in the SNV (n = 41) and SV (n = 43) history groups, when combined, represent only 1.5% (95% CI: 1.2–1.9) of the entire cohort. This indicates an extremely low prevalence of manifest mumps in the population.

All age groups were represented almost exclusively by two history groups (NSNV, NSV) in varying proportions in specific age intervals. In children’s groups, NSV exceeded 60%. Among those aged ≥ 18 years, NSNV was high (45 to 61.4%).

Taking into account the relative distribution of volunteers by history, we estimated seroprevalence in each of the specified groups (Appendix A).

The highest seroprevalence was found in those with the SNV (85.4%; 95% CI: 71.6–93.1) or SV (86.0%; 95% CI: 72.7–93.4) history, in other words, among individuals who have experience symptomatic mumps, regardless of the presence or absence of vaccination. The seroprevalence among those who denied a history of illness (NSNV, NSV) was lower: 76.5% (95% CI: 74.8–78.1) and 76.3% (95% CI: 74.8–77.8), respectively. The differences were statistically insignificant.

### 3.4. Vaccination Status

According to the National Institute of Health (Kyrgyz Ministry of Health), 20,496 people refused vaccinations in 2023 in the overall population. The reasons included doubts about immunization safety (46%), religious beliefs (43%), a lack of information about the benefits (1.3%), and other reasons (10%). False answers, or forgetfulness, during the survey cannot be ruled out either. Despite such refusals, overall population coverage with measles vaccination reached 96% in 2023. The system of prophylactic vaccination (usually with live attenuated formulations) against MMR pathogens implemented in the KR utilized several vaccines, mainly three-component M-M-R II.

In the entire cohort (n = 6617), those indicating vaccination were as follows: measles—3660 people (55.3%; 95% CI: 54.1–56.5), rubella—3201 people (48.4%; 95% CI: 47.2–49.6%), and mumps—3052 people (46.1%; 95% CI: 44.9–47.3). However, actual vaccination was only confirmed by medical documentation (date, vaccine type) for 1410 volunteers (measles), 962 volunteers (rubella), and 816 volunteers (mumps).

In most cases, a three-component M-M-R II vaccine was used (about 74% of confirmed vaccinations). The others used were Vactrivir (0.4–0.9%), bivalent vaccines for mumps-measles (2.3–4.9%) and monovalent measles, and rubella and mumps vaccines (8.8–14.4%).

## 4. Discussion

One of the indicators of the state of collective immunity is the effective reproductive number (*R_e_*), which is the number of secondary cases of infection in a heterogeneous, partially immune population [24]. By its nature, *R_e_* is the same value as *R*_0_, with the difference being that the latter indicator is valid for a completely susceptible population [25]. The most common estimates of the basic reproduction number for measles, rubella, and mumps are 12, 8, and 10, respectively [26]. Using the formula *R_e_* = (1 − (1/*R*_0_)) × 100, threshold resistance levels can be calculated, beyond which viral spread stops. For measles, rubella, and mumps, the values are 92%, 87.5%, and 90%, respectively [26,27,28].

As of November 2023, the collective immunity in the Kyrgyz population (percentage of seropositive volunteers) was 78.9% (95% CI: 77.9–79.9) for measles, 94.2% (95% CI: 93.7–94.8) for rubella, and 76.2% (95% CI: 75.1–77.2) for mumps. These results indicate that the collective immunity that existed in the KR as of November 2023 can likely prevent the epidemic spread of rubella but is not enough to prevent sporadic cases. With regard to measles and mumps, the level of collective immunity does not reach the threshold of epidemic well-being.

Regarding measles, even with the effective use of live attenuated monovalent and polyvalent vaccines, the problem of morbidity persists in most countries [29]. The main reason is considered to be a low level of collective immunity due to insufficient vaccination coverage of the population, especially in Southeast Asian and African countries. Cases of “breakthrough” infection in vaccinated healthy people, as well as in people with primary or secondary immunodeficiencies (e.g., HIV infection), have also been described [29,30,31].

As noted, given its high *R*_0_ value (12), the target measles vaccination coverage to achieve the protective immunization threshold should be about 95% [32]. The criterion for epidemic well-being for measles is the presence of no more than 7% seronegative individuals in the population. Our study, conducted at the end of 2023 in a representative volunteer cohort, showed that the share of seronegative individuals in the Kyrgyz population reached 21.0%. This is threefold higher than the acceptable level.

Taking into account that the formed cohort is representative of the Kyrgyz population, and the threshold level of collective immunity protective against measles (92–95% by WHO estimates), there is every reason to recognize the measles public health situation in the Republic as threatening [33].

This judgment was confirmed at the turn of 2023–2024, when a significant increase in measles cases was registered. As of 22 December 2023, the measles incidence in the country was 1721.3 per 100 K population according to official sources (National Scientific and Practical Center for the Control of Viral Infections, National Research Institute of Health, Kyrgyz Ministry of Health). The largest case numbers were detected among children under 4 years of age. Children under 1 year old more often fell ill in Osh city and four regions (Osh, Jalal-Abad, Issyk-Kul, Batken). Among children 1 to 4 years old, cases were most often detected in Bishkek city and three regions (Chüy, Talas, Naryn). The incidence in the child population is likely a consequence of insufficient measles seroprevalence (60–67%) confirmed in our study. Among the ill, only 7.8% of individuals were fully or partially vaccinated; the overwhelming majority had not been vaccinated against measles.

The highest collective immunity levels were observed in those over 40 years of age, wherein seronegative individuals did not exceed 7%. This may be associated with a symptomatic childhood illness, which is known to confer long-term immunity [34]. This assumption is confirmed by the fact that among these volunteers (≥40 years), individuals with high (1.01–2.0 IU/mL) or very high (>2.0 IU/mL) antibody levels predominated.

In children’s groups, where immunity was formed mainly as a result of vaccination, children with low antibody levels (0.18–0.5 IU/mL) predominated. This was also confirmed by assessment of seroprevalence by infectious and vaccinal status. In volunteers with post-infectious immunity due to symptomatic measles, seroprevalence, and Ab levels were higher than in individuals who acquired immunity through vaccination. Seroprevalence depending on activity correlated with the previously noted age dependence. The lowest levels were noted among preschoolers and schoolchildren. The highest levels were among healthcare workers and pensioners. It seems that significant seroprevalence among healthcare workers is explained by greater awareness of the importance of vaccination, as well as their involvement in additional vaccination programs as designated persons. Pensioners probably had the disease in childhood.

In the KR, measles vaccination began in 1968 (single dose at the age of 12 months). This led to a more than 40-fold decrease in incidence, but outbreaks of 8 to 10 thousand cases continued to be registered annually. In 1986, revaccination against measles was introduced at 6 years of age. This reduced the incidence to two to three cases annually, but outbreaks occurred every 4 to 5 years. In 2024, the schedule of preventive vaccinations was changed to a first dose at 12 months, with a second dose at 2 years. Since 2002, children subject to vaccination according to the national schedule reached 95–99% coverage. Additional vaccination campaigns against measles were held in different years and in various age groups: 2001 (1.8 M aged 7 to 25), 2011 (100 K aged 1–6), 2015 (374 K aged 7–20), 2018 (1.4 K aged 2–5), and 2023 (500 K aged 9 months to 7 years).

The COVID-19 pandemic has significantly affected the coverage of routine vaccination in the KR. In 2020–2021, there was a decrease in scheduled vaccination coverage, especially among children. Restrictions of movement, lockdowns, healthcare loading, as well as a growth of anti-vaccination sentiment (due to misinformation) undermined trust in immunization. As a result, the level of vaccination against measles, rubella, diphtheria, and other infections decreased nationally. This trend has continued in recent years. Low coverage is explained by the increase in the number of vaccine refusals, as well as a growth in migration (external and internal), especially in Bishkek and Osh. Children of migrants are often not registered with healthcare organizations and often miss routine vaccinations.

These data confirm the position of authorities regarding a noticeable resistance to vaccination of children in a segment of the population for various reasons, including religious factors (Department of State Sanitary and Epidemiological Surveillance, Kyrgyz Ministry of Health). It seems plausible to effectively influence such sentiments by conducting targeted explanatory work with the involvement of authoritative and informed scientists from the Muslim world. Experts from certain regions (i.e., Saudi Arabia) could be influential and beneficial insofar as the problem of vaccination hesitancy in Muslim countries is more complicated than simply the healthcare framework [35,36].

Rubella, like measles, is a vaccine-preventable infection, but despite the availability of specific vaccines, about 100,000 cases of congenital rubella syndrome occur annually globally [37,38]. The basic reproduction number (*R*_0_) for the rubella virus, according to various authors, ranges from 3 to 8, possibly reaching as high as 12 [39,40]. It follows that the required threshold of collective immunity in European countries is estimated to be 67–87%, while in developing countries, it can reach 90% [41,42,43]. Throughout the world, including the KR, a live attenuated vaccine based on the Wistar RA 27/3M strain is used for specific prevention of rubella. It is capable of preventing the infection and its most formidable complication, congenital rubella syndrome [44].

Rubella prevalence in the KR can be rightfully assessed as sporadic. As of 2023, it was 0.2 per 100 K. However, the actual incidence may be somewhat higher since this infection is characterized by a high frequency of asymptomatic cases and self-limiting courses not necessitating therapeutic intervention. It can be confidently stated that the situation is due to the high rubella seroprevalence in the population which, as indicated above, was 94.2% (95% CI: 93.7–94.8). Given that epidemic well-being implies no more than 7% seronegative individuals, isolated events of sporadic morbidity, in the context of such nearly ideal collective immunity, do not create any real grounds for public health deterioration.

The lowest seroprevalence was observed among children aged 1–17 years, 92.6% (95% CI: 91.6–93.6; *p* < 0.05). The highest was among volunteers aged 40–49 years, 97.1% (95% CI: 95.6–98.2; *p* < 0.05). No significant differences were found in seroconversion by region or activity.

Unlike the situation with measles, average anti-rubella Ab levels (25.1–100 IU/mL) significantly predominated among seropositive volunteers regardless of age or history: 47.4% in the cohort, with some variation in age groups (34.9–55.1%). Antibody levels above 100.0 IU/mL were detected mainly in children aged 1–5 years (51.1%) and in those 30 and older (>40%). This may be a consequence of previous illness. As mentioned above, rubella incidence in the Republic is currently registered to a greater extent among children aged 1–4 years.

Thus, the analysis of collective immunity to rubella in the Kyrgyz population indicates that an epidemically safe level of seroconversion (~95%) has been achieved wherein only sporadic cases are possible, mainly of imported origin.

Mumps is a widespread infectious illness that mainly affects children. In uncomplicated cases, mumps occurs as a mild, self-limiting infection. Complicated cases, including orchitis in boys, as well as pancreatitis and neurological complications, are also encountered [15,45,46,47,48,49]. Before the introduction of mass vaccination, mumps was widespread globally. In some countries, up to 5–6% of the population suffered symptomatic illness [50]. A significant change in the epidemic situation occurred after the introduction of specific vaccinations into clinical practice in 1967. This enabled a many-fold reduction in mumps prevalence throughout the world [18,51,52].

The mumps virus is less contagious than the measles virus. The average value for the basic reproduction number (R_0_) for a non-immune population, depending on the characteristics of specific regions, can vary from 4 to 7 (USA), or reach as high as 11–14 (England, Wales) [26,53]. For vaccinated individuals, more complex calculation methods are used. The SEIAR model, for example, divides the population into five categories: susceptible (S), exposed (E), infected (I), asymptomatic (A), and removed (R) [54]. Without going into the complex mathematical model used by the authors, we assume that for a heterogeneous group (with immune and non-immune subjects), the effective reproductive number R_e_ can vary from 2 to 3 [55]. Thus, according to the formula *R_e_* = (1 − (1/*R*_0_)) × 100 [28], it follows that the calculated threshold level of mumps collective immunity, which varied near ~70%, corresponded to the actual level of collective immunity in the examined cohort, 76.4% (95% CI: 75.3–77.4). An evaluation of volunteer serology showed that a high share of seropositive individuals were children aged 6–11 years (82%), and the highest was among adults 50 to 69 years (85%). The least protected were adolescents aged 12–17 years (65.7%) and adults aged 18–29 years (61.4%). In these groups, seronegative individuals exceeded 30%. It should be noted that the differences between the maximum and minimum seroconversion levels were significant (*p* < 0.05).

As with measles, mumps seroprevalence among individuals in different fields of activity corresponded with age. The highest was found among pensioners (85.6%). Likely explanations include previous contact with the virus in childhood, the frequent occurrence of asymptomatic mumps infection in adults, and an anamnestic trace in older individuals left over from the mandatory vaccination of children carried out in the USSR starting in 1980. The lowest was among students (50.6%), most of whom were aged 16–24 years. In all cases, the differences were significant (*p* ≤ 0.05). This low seroprevalence could be due to a decrease in vaccination coverage. This may relate to periods of time when such individuals should have been vaccinated according to the national schedule, yet, for various medical or administrative reasons, this was not done. An analysis of history showed that the highest share of seropositive individuals (86.0–81.1%) was observed in groups who had experienced symptomatic mumps, regardless of vaccination.

Thus, the criterion for epidemic well-being regarding mumps (≤15% seronegative individuals) was only met among adults aged 60–69 years. The rest of the population is not sufficiently protected from mumps and may be involved in epidemic processes. This includes children (all ages), adults aged 18–59 years, and the elderly (≥70 years). In general, the level of collective immunity is sufficient to prevent mass outbreaks but will not prevent isolated cases or clusters. Given the highly contagious nature of the mumps virus and decreasing immunity over time, caution is needed [56]. Adherence to vaccination schedules (primary, booster) in order to achieve the required threshold of protection will contribute to a significant reduction in incidence to, if not zero, at least low (sporadic) levels.

It is noteworthy that a significant portion of the Kyrgyz population had not been vaccinated against MMR group infections, while also not having a recorded history of illness. In children’s groups, the share of such people reached 30%. Among those over 50 years old, it approached 60%. Relatively high seroprevalence in volunteers who indicated no prior history (NSNV) indicates insufficient laboratory diagnostics of such vaccine-preventable infections. Latent and asymptomatic forms are a significant issue with this group of infectious agents [57]. Most of the older adult volunteers, born and living in the pre-vaccination period, experienced such “childhood” infections (often in mild forms). They also had the opportunity, in conditions of high morbidity, to receive a natural booster effect upon contact with infected persons.

In the KR, as in other countries, live attenuated vaccines are used for specific prevention. These can be monovalent (measles, mumps, or rubella), divalent (measles, mumps), or trivalent (M-M-R) preparations. The available vaccines are safe and effective and can be equally used within the framework of immunization programs. However, the global trend in vaccine development and prevention is a movement toward combination vaccines with more components. For routine pediatric vaccination, multicomponent vaccines are mainly used. In adults, vaccination or revaccination is carried out according to epidemiological indications using the appropriate monovalent vaccines. Vaccine type is chosen based on epidemiological conditions, including which infection is causing increased morbidity. Among the vaccinated volunteers who participated in the study, more than 70% were vaccinated with a three-component, live attenuated M-M-R vaccine.

## 5. Conclusions

1. Collective immunity to MMR group infections in the KR currently ensures epidemic well-being only with respect to rubella: seronegative individuals do not exceed 7%. For measles and mumps on average, seronegative individuals exceed 20%.

2. The highest share of individuals susceptible to MMR group infections was observed in children and young adults. Up to 40% of children were susceptible to the measles virus; post-vaccination immunity in children and young adults was characterized by low Ab levels. Approximately 30–40% of adolescents and young adults were susceptible to the mumps virus. It is likely that most volunteers of this age had never experienced these “childhood” infections symptomatically and may not have received vaccination due to parental refusal. The situation reflects existing prerequisite conditions conducive to increased incidence (measles, mumps).

3. The least susceptible to infection (MMR) were people over 50 years of age. They likely had experienced various infections (manifest or latent) with the formation of post-infectious immunity and associated high Ab levels. This age group likely also received a natural booster effect through contacts in the pre-vaccination period featuring higher viral circulation.

4. The maintenance of long-term humoral immunity is facilitated by natural booster effects, occurring in those with some degree of immunity (previously ill, vaccinated) after contact with an infected person. Obviously, the booster effect is significantly weaker in modern conditions, wherein mass vaccination has led to decreases in measles, mumps, and rubella incidence.

## 6. Limitations of the Study

The authors would like to note several factors that might affect sample representativeness or conclusions reached through data analysis. Residents who are more involved with their health and that of their loved ones (primarily women and healthcare workers) are more likely to take part in studies of this kind. The fact that healthcare workers are required to be vaccinated according to the law in many countries may also have affected the estimated seroprevalence structure in the occupational group.

Initially, information about illness and vaccination was taken from the words of volunteers and from the vaccination certificates they provided. Information about illnesses that arose in the last 15 years was confirmed by official registration data (Ministry of Health) via the iEPID digital platform. Information about vaccination was confirmed, if possible, by family doctors at the place of residence (according to medical records). The authors understand that for adult volunteers, there is a high probability that the volunteer may not remember prior illness or vaccination. In some such cases, these data could not be verified using medical records.

## Figures and Tables

**Figure 1 vaccines-13-00249-f001:**
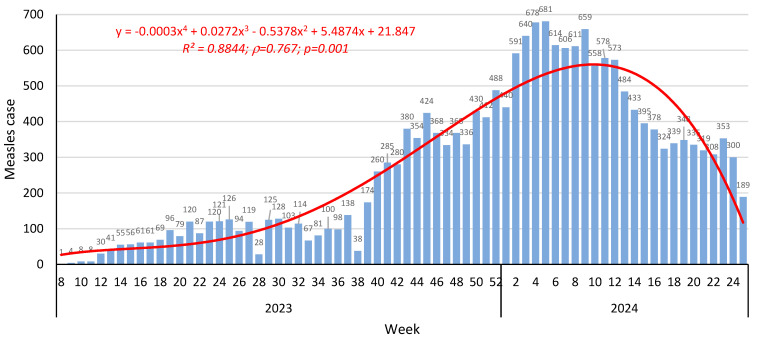
Weekly measles incidence dynamics, 2023–2024 (according to National Scientific and Practical Center for the Control of Viral Infections, National Research Institute of Health, Kyrgyz Ministry of Health). The fourth-degree polynomial trend curve is highlighted in red. Its equation, Spearman coefficient (*ρ*), and significance (*p*) are shown at the top of the diagram and highlighted in red.

**Figure 2 vaccines-13-00249-f002:**
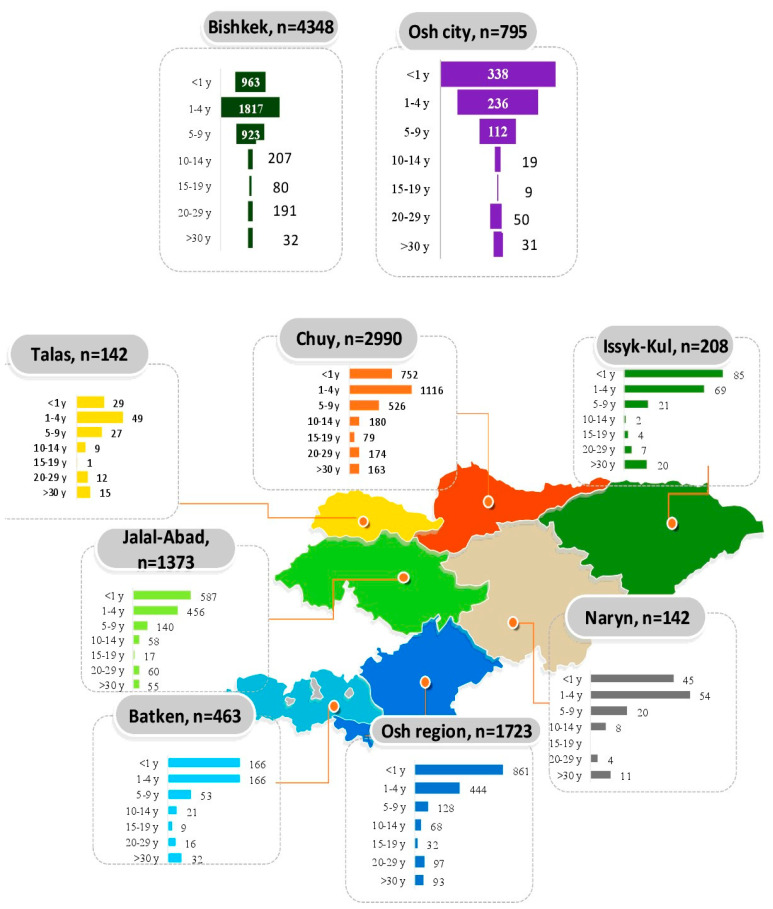
Territorial and age distribution of measles patients among persons under 30 years of age in all administrative territories of the Kyrgyz Republic as of 3 July 2023 (according to National Scientific and Practical Center for the Control of Viral Infections, National Research Institute of Health, Kyrgyz Ministry of Health).

**Figure 3 vaccines-13-00249-f003:**
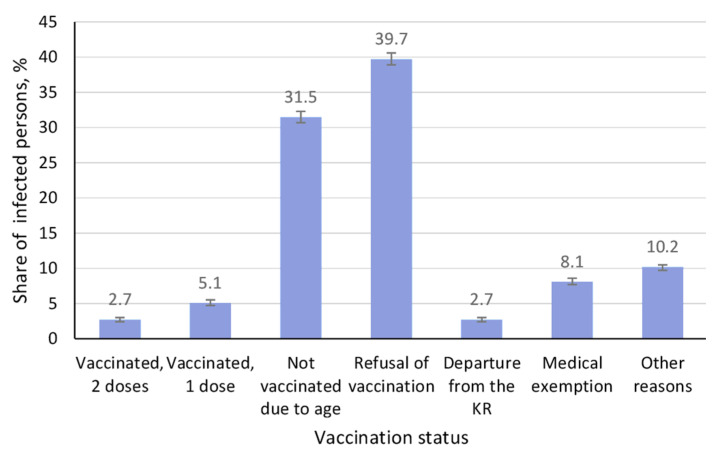
Vaccination status of residents who contracted measles in 2024. Black vertical lines are 95% confidence intervals.

**Figure 4 vaccines-13-00249-f004:**
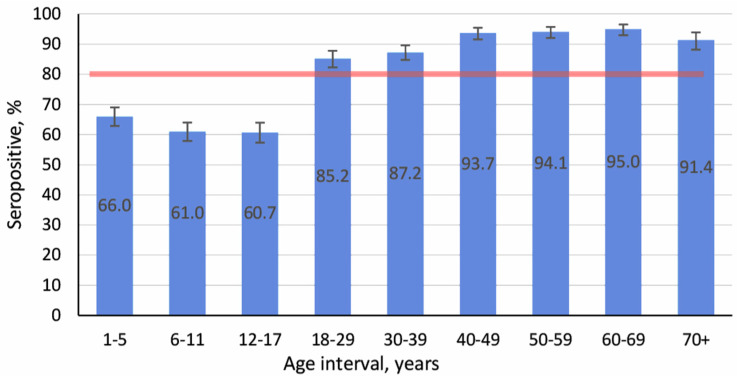
Measles seroprevalence by age. Notes: vertical black lines are confidence intervals; translucent horizontal band is the confidence interval value for the overall cohort; numerical values and significance indicators are given in Appendix A.

**Figure 5 vaccines-13-00249-f005:**
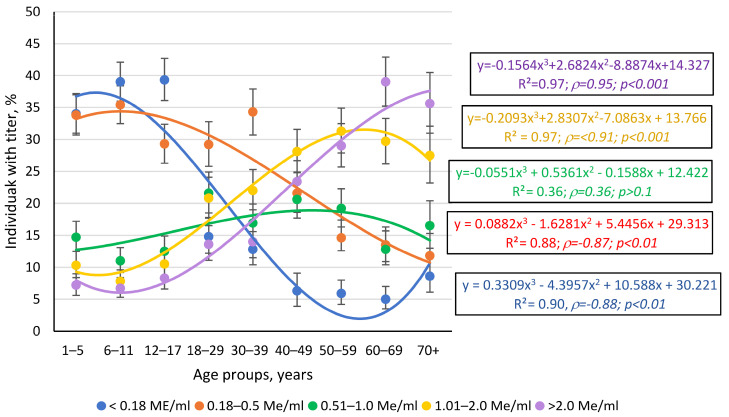
Anti-measles titer trends by age group. Legend: quantitative Ab levels, IU/mL. Boxes to the right indicate regression equation, coefficient of determination, Spearman correlation coefficient (*ρ*), and its significance (*p*). Text colors correspond to trend lines. The numerical values and statistical significance indicators are given in Appendix A.

**Figure 6 vaccines-13-00249-f006:**
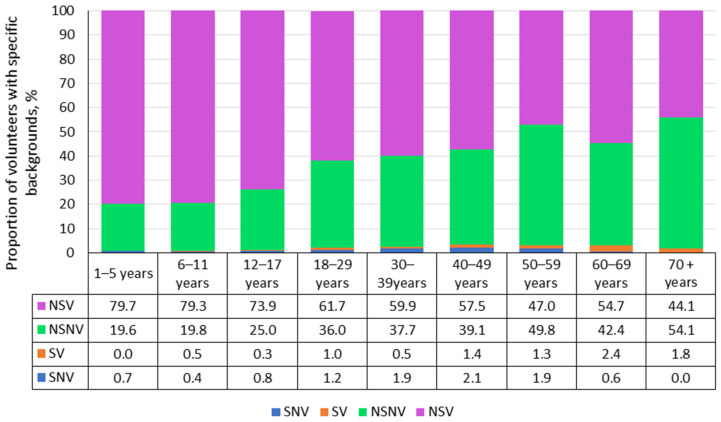
Measles history (infection, vaccination) by age group. Notes: Numerical values are given in the data table below the figure and in Appendix A. Legend: SNV—“sick, never vaccinated”, SV—“sick, vaccinated”, NSV—“never sick, vaccinated”, NSNV—“never sick, never vaccinated”.

**Figure 7 vaccines-13-00249-f007:**
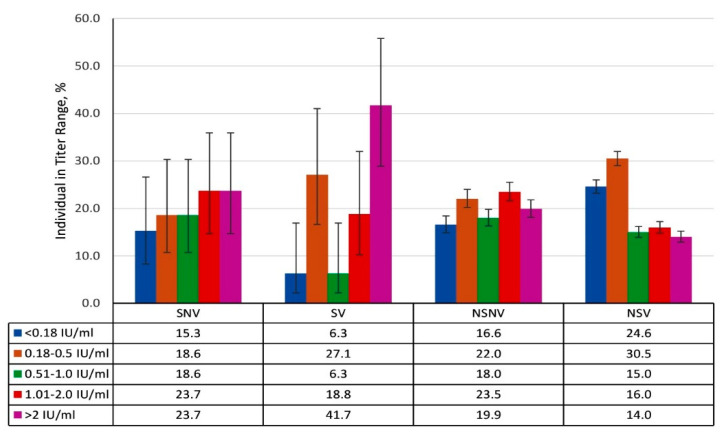
Anti-measles antibody titers by volunteer history (infection, vaccination). Notes: Y-axis: percentage of volunteers within the indicated titer range. Black vertical lines are 95% confidence intervals. Numerical values and significance indicators are given in Appendix A.

**Figure 8 vaccines-13-00249-f008:**
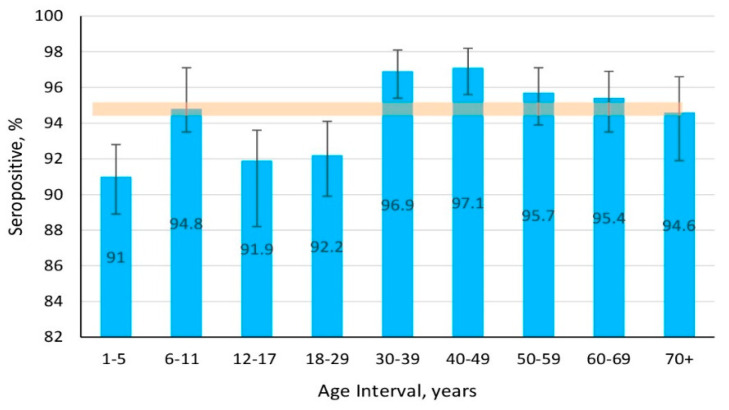
Rubella seroprevalence by age. Note: vertical black lines are confidence intervals; translucent horizontal band is the confidence interval value for the overall cohort; Numerical values and significance indicators are given in Appendix A.

**Figure 9 vaccines-13-00249-f009:**
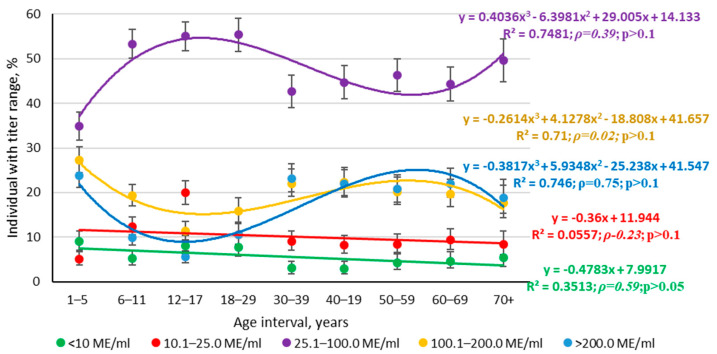
Anti-rubella titer trends by age group. Legend: quantitative Ab levels, IU/mL. Boxes to the right indicate regression equation, coefficient of determination, Spearman correlation coefficient (*ρ*), and its significance (*p*). Text colors correspond to trend lines. The numerical values and statistical significance indicators are given in Appendix A.

**Figure 10 vaccines-13-00249-f010:**
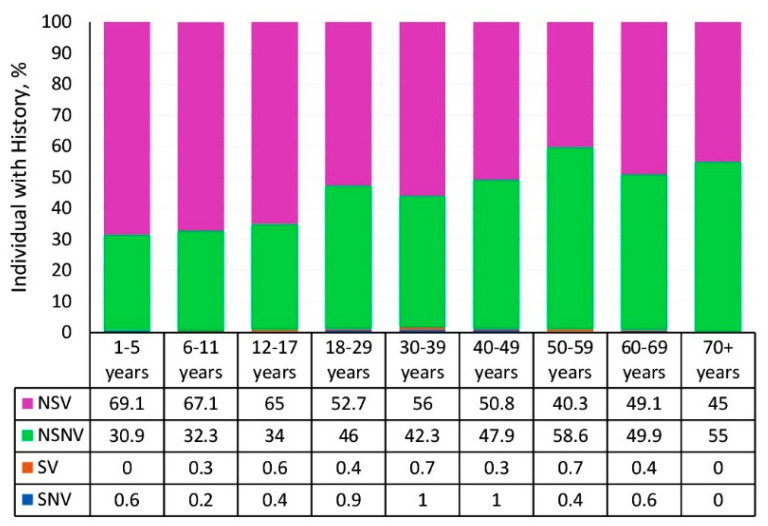
Rubella history (infection, vaccination) by age group. Note: The numerical values and significance indicators are given in Appendix A. Legend: SNV—“sick, never vaccinated”, SV—“sick, vaccinated”, NSV—“never sick, vaccinated”, NSNV—“never sick, never vaccinated”.

**Figure 11 vaccines-13-00249-f011:**
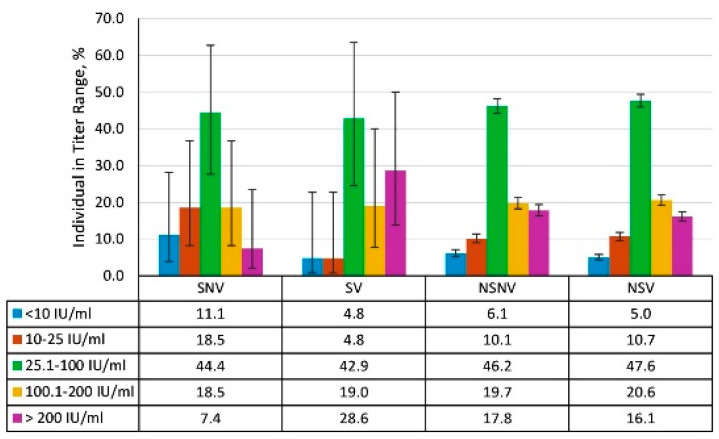
Anti-rubella antibody titers by volunteer history (infection, vaccination). Note: The numerical values and significance indicators are given in Appendix A.

**Figure 12 vaccines-13-00249-f012:**
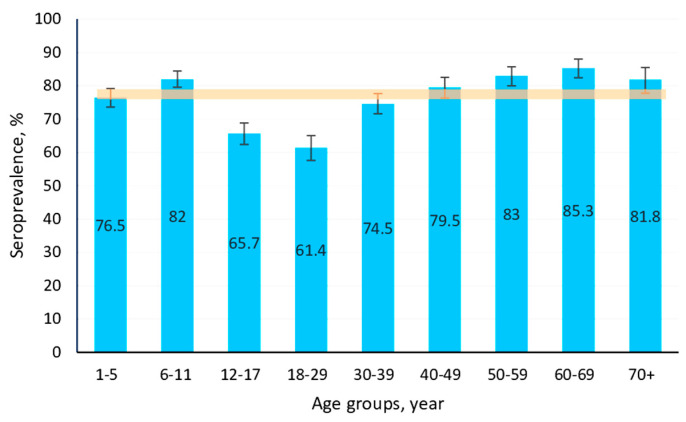
Mumps seropositivity by age. Note: vertical black lines are confidence intervals; translucent horizontal band is the confidence interval value for the overall cohort; Numerical values and significance indicators are given in Appendix A.

**Figure 13 vaccines-13-00249-f013:**
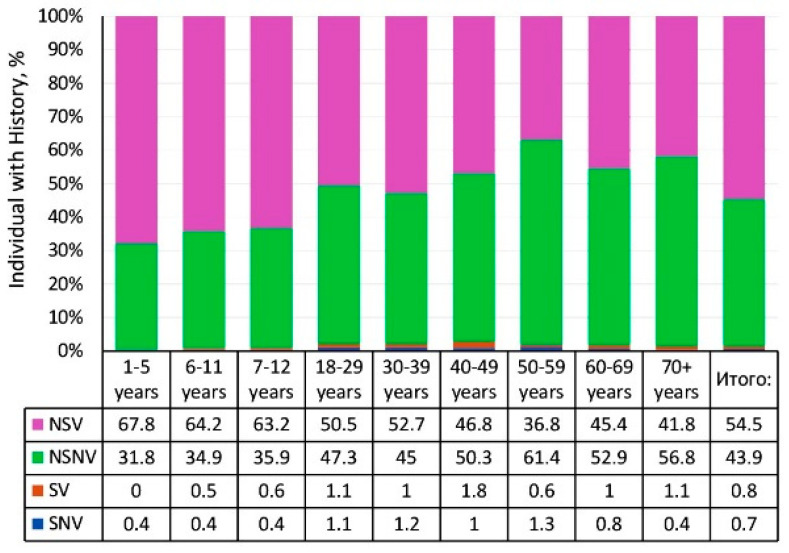
Mumps history (infection, vaccination) by age group. Note: Numerical values and significance indicators are given in Appendix A. Legend: SNV—“sick, never vaccinated”, SV—“sick, vaccinated”, NSV—“never sick, vaccinated”, NSNV—“never sick, never vaccinated”.

**Table 1 vaccines-13-00249-t001:** Age structure of the surveyed cohort.

Age Interval, Years	Individuals	Share of Cohort		Gender	
		%	95% CI	Males n (%)	Females n (%)
1–5	909	13.7	12.9–14.6	416 (45.8)	493 (54.2)
6–11	1025	15.5	14.6–16.4	460 (44.9)	565 (55.1)
12–17	877	13.3	12.5–14.1	354 (40.4)	523 (59.6)
18–29	668	10.1	9.4–10.9	136 (20.4)	532 (79.6)
30–39	686	10.4	9.7–11.1	121 (17.6)	565 (82.4)
40–49	698	10.5	9.8–11.3	107 (15.3)	591 (84.7)
50–59	692	10.5	9.8–11.2	103 (14.9)	590 (85.3)
60–69	654	9.9	9.2–10.6	107 (16.4)	547 (83.6)
70^+^	407	6.2	5.5–6.7	104 (25.6)	303 (74.4)
Total	6617	100.0	-	1908 (28.8)	4709 (71.2)

Notes: 70^+^ denotes volunteers aged 70 years and older; 95% CI—95% confidence interval.

**Table 2 vaccines-13-00249-t002:** Detailed description of the cohort by region.

City/Region	Population, Thousand	Volunteers	Share, %	95% CI	Gender	
					Males n (%)	Females n (%)
Bishkek city	1165.5	1132	17.1	16.2–18.0	343 (30.3)	789 (69.7)
Osh city	366.7	268	4.1	3.6–4.6	51 (19.0)	217 (81.0)
Osh region	1490.1	1410	21.3	20.4–22.3	392 (27.8)	1018 (72.2)
Batken region	583.4	563	8.5	7.9–9.2	170 30.2)	393 (69.8)
Jalal-Abad region	1335.8	1218	18.4	17.5–19.4	386 31.7)	832 (68.3)
Talas region	277.1	268	4.1	3.6–4.6	82 (30.6)	186 (69.4)
Issyk-Kul region	544.4	538	8.1	7.5–8.8	171(31.8)	367 (68.2)
Naryn region	312.1	339	5.1	4.6–5.7	83 (24.5)	256 (75.5)
Chüy region	1086.8	881	13.3	12.5–14.2	230 (26.1)	651 (73.9)
Total	7169.9	6617	100	-	1908 (28.8)	4709 (71.2)

Note: as of 2024. Source: https://stat.gov.kg/ru/opendata/category/39/, accessed on 16 January 2025.

**Table 3 vaccines-13-00249-t003:** Distribution of volunteers by activity.

Activity	Individuals	Share, %	95% CI
Preschooler	648	9.8	9.1–10.5
Schoolchild	1632	24.7	23.6–25.7
Student	164	2.5	2.1–2.9
Medicine	1276	19.3	18.4–20.3
Science + the Arts	47	0.7	0.5–0.9
Business	86	1.3	1.1–1.6
Education	198	3.0	2.6–3.4
Production + Transport	51	0.8	0.6–1.0
State-military Service	184	2.8	2.4–3.2
Office Work	81	1.2	1.0–1.5
Info. Technology (IT)	66	1.0	1.8–1.3
Agriculture	173	2.6	2.2–3.0
Other	668	10.1	9.3–10.8
Unemployed	731	11.1	10.3–11.8
Pensioner	612	9.2	8.6–10.0
Total	6617	100	-

**Table 4 vaccines-13-00249-t004:** Rubella incidence for 11 months of 2023.

City/Region	Cases		Per 100 K Population	
	Overall	Among Children 1–14 Years	Overall	Among Children 1–14 Years
Bishkek city	5	5	0.4	2.2
Osh city	1	1	0.3	1.0
Batken region	0	0	0	0
Jalal-Abad region	2	2	0.2	0.4
Issyk-Kul region	0	0	0	0
Naryn region	0	0	0	0
Osh region	0	0	0	0
Talas region	0	0	0	0
Chüy region	3	3	0.3	0.9
Total	11	11	0.2	0.5

**Table 5 vaccines-13-00249-t005:** Mumps incidence for 11 months of 2023. Source: Official data from the Department of State Sanitary and Epidemiological Surveillance.

City/Region	Cases		Per 100 K Population	
	Overall	Among Children 1–14 Years	Overall	Among Children 1–14 Years
Bishkek city	30	17	2.6	7.4
Osh city	7	2	1.9	1.9
Batken region	27	19	4.7	8.8
Jalal-Abad region	18	13	1.4	2.7
Issyk-Kul region	4	3	0.7	1.7
Naryn region	1	1	0.3	1
Osh region	8	4	0.5	0.7
Talas region	7	7	2.5	7.1
Chüy region	59	51	5.5	15.2
Total	162	117	2.3	5.1

## Data Availability

Data are contained within the article.

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
