# Peer review of "Collective Immunity to the Measles, Mumps, and Rubella Viruses in the Kyrgyz Population"

_vaccines, 2025, doi:10.3390/vaccines13030249_

Round 1
Reviewer 1 Report
Comments and Suggestions for Authors
This manuscript by Popova et al describes a nationwide survey for population immunity against measles, rubella, and mumps in the Kyrgyz Republic. It was shown that the seroprevalence against rubella is sufficient to interrupt the outbreak in all age groups, while the seroprevalences against measles and mumps are inadequate, especially in children. This explains well why a measles outbreak occurred among children in the Kyrgyz Republic in 2023-2024. This study will not only provide crucial information for vaccine policy making in the Kyrgyz Republic but will also contribute to the international measles and rubella elimination efforts.
To improve the manuscript, please refer to the following comments.
1) This manuscript is very redundant; thus, it should be rewritten to be short and concise. Some tables and figures, for example Table 1 and Figs 1 and 2, share the same data and should either be deleted or moved to Supplementary Information. Another reason for the long manuscript is the overlapping of descriptions in the Results and Discussion sections.
2) Is there any information on the history of routine and supplementary immunization schedules for measles, rubella and mumps in the Kyrgyz Republic? The collective immunity data in this study may be better understood in depth if it is known which age groups had the opportunity to be vaccinated.
3) 2.1. Characteristics of the surveyed volunteer cohort
Why is the percentage of women very high among adults in the surveyed volunteer cohort? Does this reflect the gender ratio of the population of the Kyrgyz Republic? If not, please discuss the implications for the current survey.
4) 2.2. Research methods
Please describe the positive and negative cut-off values for the ELISA kits used. It would be helpful for the reader to understand the plausibility of the present results, especially since the measles and rubella IgG assay kits are presented in international units. Are there “Equivocal” or “gray zone” results in these kits? If so, please indicate how you handle such results.
5) 3.1.1., 3.2.1., 3.3.1. Epidemiological situation
Please provide data sources on the epidemiological situation for measles, rubella, and mumps, including the number of reported cases, geographic distribution, and vaccine status of the cases.
6) The key point and the biggest question in this manuscript is that there are large differences in antibody prevalence among measles, rubella, and mumps.
The authors describe in lines 461-464 that “Despite such refusals, overall population coverage with measles vaccination reached 96% in 2023. In most cases, three-component MMR vaccines (Vactrivir, M-M-R II vaccine) were used.” As shown in Figures 13, 19, and 25, in the present study cohort as well, the highest proportion of volunteers who were confirmed to have received vaccination also received the trivalent vaccine. As the authors cite in the introduction, the Cochrane Review describes the efficacy of the measles, rubella, and mumps vaccines as 95%, 89%, and 72%, respectively.
For the 1-5-year-old group in the current study cohort, if there was a 96% immunization coverage and trivalent vaccine was used for most of them, the results of the antibody prevalence would be reasonable for rubella (94.7%) and mumps (76.5%), but clearly low for measles (66.0%). It is necessary to carefully discuss the reasons for the low antibody prevalence against measles. Since the NSNV history rates are lowest for measles as shown in Figs.11, 17 and 23, self-reported vaccination history is considered insufficient to explain the results.
Similar data were also reported by Hachiya et al. (doi.org/10.1371/journal.pone.0194931) and may be helpful for the discussion.
Minor points
1) Please review all references to make sure they are all properly cited. At least reference 16 for lines 88-92 and reference 11 for lines 93-94 seem wrong.
2) Lines 95-98, It would be more appropriate to change “seropositivity” to “vaccine efficacy”.
3) Lines 258-259, “The total contribution of other volunteers to prevalence was 12.3% (95% CI: 11.4-12.6).” Age group” seems appropriate, not ‘volunteers’.
4) Table 2. What is the unit for “Population” column? There are decimal points, so it must be not a simple “number”.
5) Line 365, Change “Table 4e” to “Teble 4s”.
6) “Funding: The study was carried out entirely with government funding.” Please state which government funding.
Reviewer 2 Report
Comments and Suggestions for Authors
Too long - and too many figures - focus on main issues, and quality not quantity of data
Need to describe background - eg of vaccination policies and coverage over time and of disease notification practices and numbers over time. How sensitive and accurate are vaccination and disease reporting likely to be ? Discuss critically.
No mention of hospitalisations, or deaths, or eg congenital rubella syndrome
Sample not representative - eg volunteers and huge excess of adult females - needs critical discussion. What is likely direction of any bias ?
Herd ("collective") immunity thresholds - assume solid lifelong immunity and random mixing !
Need to stratify data by age and discuss eg Figs 6 and 12. Fancy polynomial curves eg on Figure 10 can be omitted
Need critical discussion of validity of disease and vaccination histories - these data are likely to be very weak - are they worth presenting ?
Discuss quality control of the serology
Discuss recent trends in coverage and vaccine hesitancy, as this is a widespread problem. Maybe mention surrounding countries.
Comments on the Quality of English LanguageEnglish pretty good, but word "prevalence" is misused repeatedly in place of incidence, and sometimes "incidence" or "infection" is used when should say reported cases. Probably wise to have paper checked by native English speaker.
Reviewer 3 Report
Comments and Suggestions for Authors
Thank you for the opportunity to review manuscript ID: vaccines-3483591. This manuscript aimed to investigate the level of collective immunity to the measles, mumps and rubella (MMR) viruses in the Kyrgyz Republic.
Comments:
Line 78: Add a new paragraph in which the data about the incidence/mortality of measles, rubella and mumps should be described in the previous period in the Kyrgyz Republic, citing appropriate references. Describe the trends in the frequency of these diseases, as well as the epidemics of these diseases (with a description of the age groups that were at risk the most).
Line 83: Add a new paragraph in which a brief history of MMR vaccination in the Kyrgyz Republic should be described (including the years of introduction of these vaccines in mandatory immunization programs, vaccination schedule, age covered by vaccination, coverage of vaccinations, whether they were paid for or free), citing relevant references. Also, indicate whether the supply and availability of these vaccines was continuous or if there were any interruptions in the supply of these vaccines (and in what period). Were there activities such as catch-up immunization for MMR in the previous period in the Kyrgyz Republic.
Lines 140-153: Explain the disproportionate distribution by gender in all age groups. It is difficult to assume that the distribution by gender in this study is consistent with the distribution by gender in the Kyrgyz Republic. What caused such a distribution by gender?
Line 206: Add a new paragraph in which the definitions for outcomes should be given, that is, for persons who are considered seropositive or seronegative for measles, mumps and rubella, citing appropriate references.
Lines 271-297: Describe the data presented in Figure 6 by age.
Lines 318-327: It is not good practice to cite references from other authors in the Results section. Correct this so that such a comparison is carried out in the Discussion section. In the same way, check and correct the Results section as a whole.
In Supplementary data Table 12s is missing. Correct this.
Check Table 10s.
Line 939: Add a new paragraph to state and discuss the limitations of this work (such as potential sources of bias such as disproportionate selection by gender, variable `Medicine` by activity, etc.). Discuss the possibilities for eliminating the limitations of this work.
Round 2
Reviewer 3 Report
Comments and Suggestions for Authors
Thank you for the opportunity to re-review manuscript ID: vaccines-3483591.
The authors have addressed all of the issues highlighted in my review and responded in a quality manner to my questions. Also, the authors made the necessary changes to the manuscript. Thank you to the authors.